# Imaging Microstructure on Optically Rough Surfaces Using Spatially Resolved Acoustic Spectroscopy

**Wenqi Li** [1,*,†], **Paul Dryburgh** [2,†], **Don Pieris** [3], **Rikesh Patel** [1], **Matt Clark** [1] **and Richard J. Smith** [1]

1. Optics and Photonics Group, University of Nottingham, Nottingham NG7 2RD, UK
2. Department of Surgical & Interventional Engineering, School of Biomedical Engineering & Imaging Sciences, King's College London, London SE1 7EH, UK
3. Centre for Ultrasonic Engineering, University of Strathclyde, Glasgow G1 1XQ, UK
* Correspondence: wenqi.li@nottingham.ac.uk
† These authors contributed equally to this work.

**Featured Application: Measurement of material microstructure on industrially relevant surface finishes.**

**Abstract:** The microstructure of a material defines many of its mechanical properties. Tracking the microstructure of parts during their manufacturing is needed to ensure the designed performance can be obtained, especially for additively manufactured parts. Measuring the microstructure non-destructively on real parts is challenging for optical techniques such as laser ultrasound, as the optically rough surface impacts the ability to generate and detect acoustic waves. Spatially resolved acoustic spectroscopy can be used to measure the microstructure, and this paper presents the capability on a range of surface finishes. We discuss how to describe 'roughness' and how this influences the measurements. We demonstrate that measurements can be made on surfaces with $R_a$ up to 28 μm for a selection of roughness comparators. Velocity images on a range of real surface finishes, including machined, etched, and additively manufactured finishes in an as-deposited state, are presented. We conclude that the $R_a$ is a poor descriptor for the ability to perform measurements as the correlation length of the roughness has a large impact on the ability to detected the surface waves. Despite this issue, a wide range of real industrially relevant surface conditions can be measured.

**Keywords:** SRAS; rough surfaces; laser ultrasound; microstructure imaging; orientation imaging

## 1. Introduction

Most materials used in high-value engineering applications are chosen for their mechanical performance. The material is optimised in terms of the chemical composition and the processing routes used to achieve the desired microstructure to deliver the performance needed by the part when it is in service. Knowing the material state is challenging as there are a large number of processing steps performed to produce the final part, and they are performed on the raw material, the alloyed billet, heat treatments pre- and post-machining; all of these steps can change the material state and so influence the performance of the part.

For additive manufacturing (AM), the challenges are even more pronounced, as the material state is formed during the manufacturing process, which is inherently hard to control and so traditional approaches for quality control and assurance are difficult to apply. There are a number of different AM processes that have very different characteristics in terms of the influence on the material state, the range and type of defects, and the surface finish achieved. They can be categorized by their material feed-stock, e.g., wire, blown powder, or powder bed, and their method of energy input, e.g., arc, e-beam, and laser. Aside from traditionally manufactured machined and etched samples, this paper looks at samples from two AM methods, wire additive manufacturing and laser powder bed fusion, and details of the samples used are given in Section 2.3. Wire arc additive manufacturing

(WAAM) uses wire feed-stock and an electrical arc to melt and weld the new material to the previously built layers [1]. Wire laser AM (WLAM) is similar to WAAM except that the wire feedstock is melted by a high-power laser beam [2]. They have relatively fast deposition rates, with rates in excess of 10 kg/h possible; however, this comes at the cost of resolution and part complexity. Laser powder bed fusion (L-PBF) uses a laser to melt metal powder a layer at a time to build the part. This has high spatial resolution, allowing for very complex geometries to be built, but with smaller build volumes and lower rates of deposition compared to WAAM < 200 g/h [3].

A multitude of defect mechanisms threaten the integrity of a build, which can include obvious geometrical errors to the more subtle internal porosity [4], micro-cracking [1], and heterogeneous microstructure formation [5].

The ability to determine the material state is a limiting factor in the application of these manufacturing methods to safety-critical parts despite the wealth of advantages they have over traditional manufacturing methods.

It is useful to break the types of material non-conformity down into two categories, defects and microstructural. The first is concerned with physical defects both through the volume and on the surface, for example, cracking and porosity. For the detection of such issues, there is a mature non-destructive evaluation (NDE) industry, utilising tools including eddy current [6] and dye penetrant [7] for surface defects, whilst internal defects and porosity are inspected by X-ray [8] and conventional ultrasound.

Processing steps for both traditional and additive manufacturing affect the microstructure of the final part and can cause deviations between the intended crystalline microstructure and that of the real part. Common microstructural non-conformities include grain size, texturing, and the presence of microtextured regions, all of which act to reduce the performance of the part. Traditional statistical qualification processes for metallic materials are well defined [9], requiring extensive testing, which can take several years to complete with costs running into the millions [10]. This is typically based around mechanical testing to establish the components yield and ultimate tensile strength [11] and is performed over a large number of specimens to obtain high confidence in performance across each build set. For example, small sections of material are prepared and polished, and the microstructure is imaged with electron backscatter diffraction (EBSD) [12]. Clearly, this raises two key issues; firstly, additive manufacturing has many benefits that make it especially attractive when low production runs are required, even producing one-off components [13], making established qualification regimes an unsuitable approach [14]. Turning this problem on its head also presents a significant opportunity; if the material that sees service, be it machined or additively produced, can be characterised, there is an opportunity to realise a new paradigm in material efficiency, removing the need for material design tolerances to ensure safety.

Spatially resolved acoustic spectroscopy (SRAS) [15,16] is a laser ultrasound technique using surface acoustic waves (SAWs) to measure material microstructure. SAWs are generated with a fixed wavelength, meaning the waves that propagate have a frequency determined by the acoustic velocity of the material at the generation location. As this does not rely on the time of flight, this produces a simple and robust measurement.

Measuring the velocity allows spatial maps of the material microstructure to be obtained as the SAW phase velocity varies with the crystallographic orientation for most materials. Beyond grain maps, if more measurements for a range of SAW propagation directions are taken, then the crystal orientation [17,18], and more recently, the single crystal elasticity matrix [19], can also be obtained. SRAS has been applied to a range of engineering materials and their alloys, e.g., titanium, nickel, steel, and silicon [18,20,21]. Samples are typically prepared before scanning, for example, sliced, ground flat, or lightly polished. The smooth surface is currently required as the knife edge detector (KED) that is usually used only works with specular reflections (see Sections 2.1 and 2.2). While this preparation is acceptable for research specimens, the preparation of real parts is undesirable as it affects their function.

Translating SRAS measurements from prepared and polished surfaces to those of real parts introduces a number of new challenges. These can impact the form of the instrument (curved surfaces require complex sample handling), while others have a more fundamental impact on how the measurement can be performed. This includes the impact of the optically rough surface found on most real parts. The roughness is determined by how the material was made (if additively manufactured) or how the part was machined and the treatments it might have experienced. The scale of the roughness determines the challenge it imposes for both the generation and propagation of ultrasound, as well as the impact on the detection laser beam. This will be discussed in more detail in Section 2.1, but briefly, roughness of the surface can modify the ultrasound generation process by changing the effective wavelength and thereby reduce the efficiency of the generation process. Extremely rough surfaces can have additional effects on the measurement; they may increase the attenuation, change the apparent velocity, or interfere with the ultrasound generation process [22]. For the detection process, the main effect is the scattering of the detection laser beam, not only reducing the light return to the detector but also producing optical speckles, which not all detection schemes can tolerate.

This paper presents progress on using SRAS on optically rough surfaces for machined, etched, and additively manufactured parts (via WAAM, WLAM, and L-PBF). The challenges and limitations imposed by the surface on the spatial and velocity resolution, as well as the impact on scan speed, are also discussed.

## 2. Materials and Methods

This section of the paper describes the roughness found on different types of surfaces, introduces ways to categorise these surfaces, and considers the impact they have on ultrasound measurements. The SRAS instrument used for taking measurements is introduced, and the materials and samples presented are described.

### 2.1. What Is 'Roughness'

The roughness of a surface depends upon the height variations and the spatial correlation of those variations. As an example, when considering the surface of sample, three distinct roughness domains can be identified: *form*, semi-periodic mesoscopic form or *waviness*, and (optical-scale) *roughness*. These are separated using different cut-off lengths, also referred to as the correlation length [23]. For the purposes of this work, this distinction is useful as each regime presents a different challenge to laser ultrasound measurements.

Figure 1a shows how the surface influences the probe laser beam. When the surface is smooth, the *form* is essentially flat with respect to the detection, and so the laser returns to the detector; if the surface is *wavy*, the local gradient means the returning light is reflected at a different angle when the sample moves, which means that it can return outside of the acceptance angle of the collection optics and be lost. For small-scale *roughness*, the light is scattered over a wide range of angles.

The changes to the return probe light cause problems for the knife edge detector used in the instrument; Figure 1b shows the ideal case, where the laser reflects from the smooth surface and returns to the detector, where both photodiodes are illuminated equally, allowing balanced detection and cancellation of common mode noise, which produces optimal signals. In this case, the primary challenge is to maintain the focus of the optical system on the specimen surface. Figure 1c shows the influence of a surface with waviness, where the surface normal is changing quickly moving the reflected spot off-centre, away from the optimal balanced position, and in the worst cases reflecting light out of the acceptance angle of the optical system, causing signal dropouts. In Figure 1d, the return light goes everywhere, but speckles are also introduced, which are across both photodiodes, and so when the speckles move due to the presence of the sound, both light and dark areas move across the knife edge and no signal is obtained.

The dataset shown in Figure 1e was acquired using Mitutoyo-Surftest SV-600 Profilomet [24], whereby a tool is used to measure the variation in height across the surface

of the specimen; this can be performed mechanically or optically. From a mechanical perspective, one can use surface profilometry, where a stylus is dragged across the surface of the inspected component [25]. From an optical perspective, a wide variety of microscopy and interferometry methods can be used, including focus variation and confocal and coherence scanning [26]. The data produced by these techniques then allow the extraction of representative roughness parameters. A range of such parameters are defined by ISO 21920, including $R_a$, the arithmetic mean deviation of the measured profile; $R_q$, the arithmetic mean deviation of the measured profile; and $R_z$, the arithmetic mean deviation of the measured profile [27]. Of these, $R_a$ is the most commonly used and is therefore used throughout this work. Where the $R_x$ parameters are concerned with describing roughness over a line profile, the $S_x$ describes the areal roughness; however, these are yet to see widespread industrial use [28].

It is important to convey that the measured roughness is highly dependent on the tool used, and the length scale across which it operates [29]; coordinate measuring machines, for example, are generally used to capture features from the millimetre to metre scale, whilst on the opposite end of the spectrum, atomic force microscopy is concerned with measurements on the micron scale. Therefore, to elucidate a greater understanding of surface roughness, it is often necessary to fuse measurements across length scales.

The notion of spatial frequency is essential in surface metrology; the calculation of descriptive roughness parameters requires the filtering of low-frequency components (to remove form and waviness) and is often referred to as multi-scale analysis [30]. An example of this is given in Figure 1e, where the unfiltered surface profile of this sample is shown using the black line. A cut-off length of 1 mm is used here to extract the surface form (blue line). The next cut-off length of 0.025 mm was used to separate the waviness (green line) from the surface roughness (red line). The correlation length used is in line with ISO 4288 for these types of surfaces and is discussed to a greater detail in the literature [23].

Beyond being used as a cut-off length, the correlation length has a further importance in acoustics for describing the attenuation of the SAW due to the surface roughness. The attenuation and frequency shift of SAWs have been explored at length by Maradudin and various collaborators [31], primarily through perturbation theory. This body of work concludes with two primary findings, namely thatSAWs propagating across a rough surface are subject to (1) attenuation and (2) dispersion, and that the degree to which these effects occur has a complex dependency on the acoustic wavelength, surface height, and spatial frequency. For example, in the Rayleigh regime, attenuation varies as the fourth power of the frequency, suggesting that for high-frequency surface wave propagation, the surface roughness may play a critical role [31]. Ruiz and Nagy have previously measured these effects in machined aluminium specimens by laser ultrasound [32], but the extent to which this effect is observed in additive manufacturing samples is yet to be studied. Conversely, there has been shown to be an improvement in generation on rough surfaces. Bakre et al. demonstrated [33] that the amplitude of a SAW generated by laser-excitation increased with increasing roughness. Bergström et al. later demonstrated by ray-tracing that this effect was due to multiple scattering and increased with increasing roughness [34], with it being particularly pronounced in high-reflective metals.

Whilst the $R_a$ value is widely used to describe the roughness of the surface, when comparing a variety of surfaces of similar $R_a$, it quickly becomes clear that many different types of surface can present with a similar $R_a$. This is discussed in more detail in Section 2.3.1.

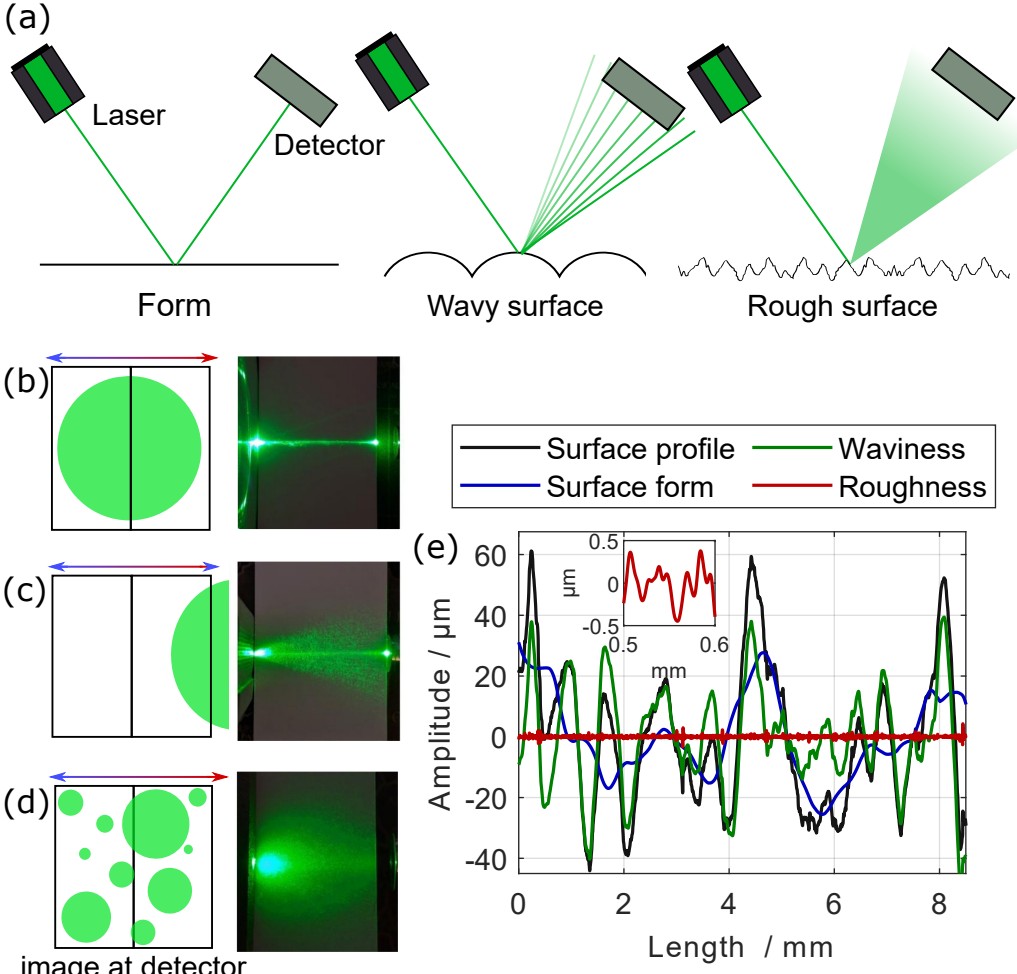

**Figure 1.** (**a**) Diagrams illustrating how light reflects off a smooth surface where the *form* is such that it appears locally flat to the detection (left), a *wavy* surface (middle), and a *rough* surface (right). (**b**) diagram showing the centred beam returned to the detector from the smooth surface reflection, and a photographs of a green laser reflected off the surface of an optically smooth surface (left). (**c**) shows the beam reflecting at a different angle from a wavy surface and photograph showing the light returned from a WAAM sample. (**d**) shows the speckles on the detector from the fine-scale rough surface, and the L-PBF sample shows the effect on the detection laser. (**e**) decomposition of surface profile (black) of an as-deposited L-PBF specimen into the three regimes of roughness: form (blue), waviness (green), and roughness (red). The inset figure shows a zoom of the roughness value across of length of 100 μm—on the order of the diameter of the detection spot size of the rough surface SRAS instrument.

## 2.2. SRAS Instrumentation for Optically Rough Surfaces

SRAS uses two lasers for the non-contact generation and detection of acoustic waves; the basic optical diagram of an SRAS instrument is given in Figure 2a. Considering first the generation arm of the system, a pulsed near infrared laser (NIR) illuminates an amplitude grating of period $M\lambda_g$ (1:1 mark space ratio). This is then re-imaged onto the sample surface with magnification of M, as determined by the ratio of the two lenses used, producing an image of the grating on the sample surface with a period $\lambda_g$. Through thermo-elastic absorption of the pulsed laser energy, acoustic waves are generated in the specimen.

A schematic of the image formed on the sample surface is shown in Figure 2b, where the distance between the fringes of light correspond to the wavelength of the generated SAW, $\lambda_g$. Therefore, the SAW velocity for a given point can then be found by $v_{SAW} = f\lambda_g$.

The generation lasers used have pulse energies from 800 μJ–4 mJ, with repetition rates from 1 to 5 kHz.

The detection beam is shown in green; it is important to note that the frequency, $f$, is determined by the material under the generation patch, it is non-dispersive and so does not vary with propagation distance, and thus, the SAW velocity recovered is the same regardless of the detection beam being placed at location (i) (within the same grain as the generation) or (ii) (neighbouring dissimilar grain). This makes for a very robust measurement system where the signal amplitude, generation and detection efficiency, and acoustic aberrations do not affect the measurement. Importantly for rough surfaces, this allows the detection beam to be positioned very close (~1–3 wavelengths) to the edge of the generation patch, minimising the effects of attenuation.

SRAS instruments make use of broadband Q-switched lasers for the generation of ultrasound, with pulse widths in the range of 1–10 ns. These short pulse widths allow access to acoustic waves of frequencies from the low MHz to the high hundreds of MHz, which allows flexibility in the bandwidth of the detector used.

The principle of the SRAS technique is agnostic to the type of detector system used. By way of example, Figure 2c–f present SRAS velocity maps captured using a number of different detector systems. An optical image is also given—so that the surface condition of the samples can be seen—along with a photograph of the detector head. The KED (c) is used for prepared-mirror-like surfaces using a split photodiode for balanced detection and noise cancellation.

Blum et al. have previously introduced and demonstrated the quadrature demodulation technique for the measurement of acoustic waves [35], and systems using this approach are now available commercially (the example in Figure 2d uses the fibre-coupled Quartet manufactured by Sound and Bright).

The speckle knife-edge detector (SKED) is an evolution of the KED, which was developed at the University of Nottingham to detect acoustic waves from optically rough surfaces [36]. To cope with the speckle, an array of photodetectors are used where the electronics can identify the individual points of speckle and split them to form a knife edge on each speckle to produce a right- and left-channel signal as found in the smooth surface KED.

Laser Doppler vibrometers are widely used in both academia and industry, with many turn-key commercial systems available (the example in Figure 2f uses a Polytec OVF-5000).

Other types of optical ultrasound detectors are available commercially, for example, those that use two wave-mixing [37,38] or Fabry Perot cavities [39]. These techniques work well on optically rough surfaces but have not to date been integrated into an SRAS instrument.

This paper presents results in the form of SAW velocity images from a single direction of propagation. The maps shown in Figure 2c–f show that for the SRAS technique, the choice of the detector does not influence the measurement of the SAW velocity. Typically, the detector is combined with the SRAS optics so that the detector spot size and stand-off distance are optimised for the generation process.

The main results of the paper in Section 3 were performed with the Quartet detector, as this allows the impact of a range of surface finishes on the performance of the generation and detection processes to be compared. Some industrial samples are presented from the SKED detector to show that these rough surfaces can be measured using other detection techniques.

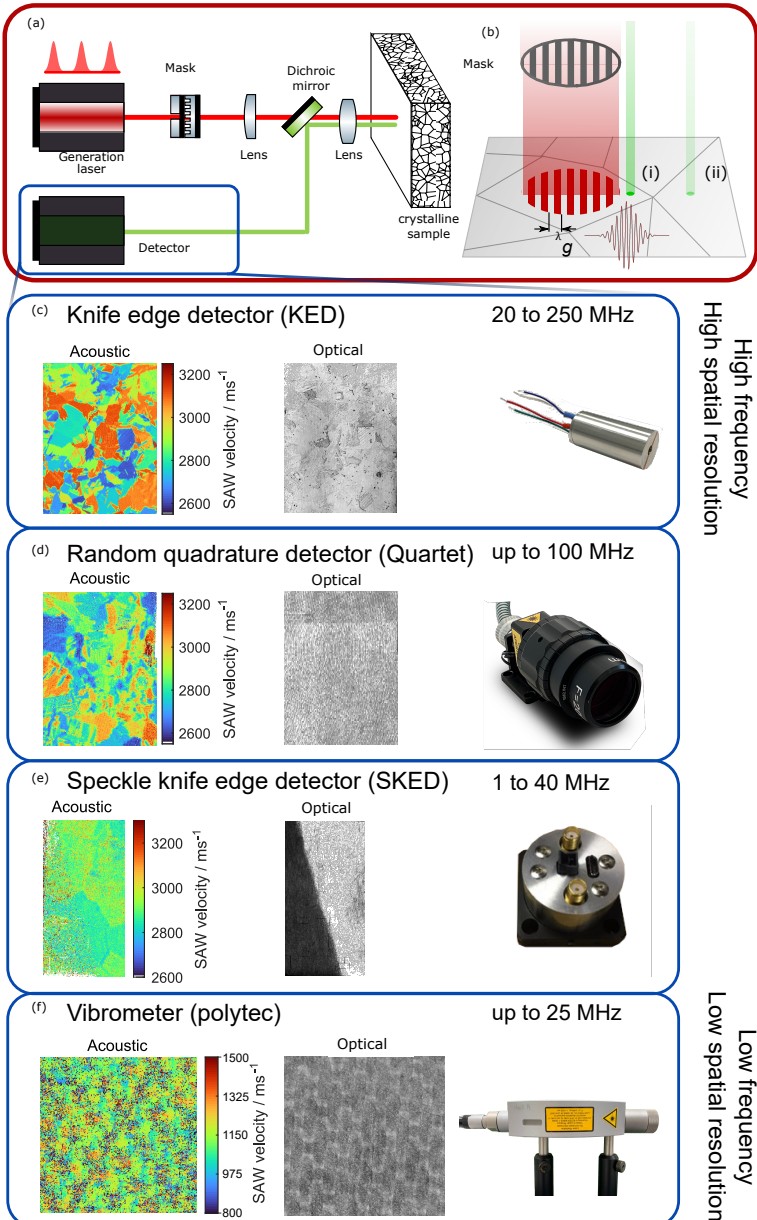

**Figure 2.** SRAS can be used with a variety of detectors, depending on the sample and surface finishing to be interrogated. (**a**) shows generalised schematic of the optical system and sample to be scanned, where the detector may take the form of a commercial system with integrated laser and detector, or a bespoke configuration separating the input laser and detector into two arms. (**b**) shows the area of interest on the sample surface, where the acoustic waves are generated under the imaged mask pattern and have wavelength $\lambda_g$; an example of the generated waveform is then shown propagating towards the detection position. As SRAS measures the frequency of the wavepacket—not the time of flight—the velocities determined at positions (i) and (ii) are the same, despite crossing a grain boundary. (**c**–**f**) then give examples of previous SRAS scans using different laser ultrasound detectors.

### 2.3. Materials and Samples

Table 1 lists the samples presented in the results section, detailing the material, the type of surface finish, the detector used, and the spatial resolution of the instrument. Section 2.2 introduces the SRAS technique and a range of detectors that have been used as part of SRAS instruments in the past. To allow us to draw general conclusions on the influence of the surface conditions, the bulk of the measurements are performed with the Quartet

detector. The rest of this section gives more details on the samples with respect to how they were prepared and scanned.

**Table 1.** Overview of samples.

| Sample | Figure No. | Material | Surface Finish | Detector | Spatial Resolution μm |
|---|---|---|---|---|---|
| Comparators | Figures 4 and 5 | Nickel | Various see Table 2 | Quartet | 400 |
| WAAM 1 | Figure 7 | Ti-6Al-4V | As deposited | Quartet | 500 |
| Ti-6246 | Figure 8 | Ti-6Al-2Sn-4Zr-6Mo-Si | Etched | Quartet | 400 |
| L-PBF 1 | Figure 9a | Ti-6Al-4V | As deposited | SKED | 500 |
| L-PBF 2 | Figure 9b | AlSi10Mg | As deposited | Quartet | 500 |
| CPTi | Figure 10 | Titanium | Various | Quartet | 400 |
| Ti64 | Figure 11 | Ti-6Al-4V | Milled | SKED | 500 |
| WLAM | Figure 12a | TiB2 | As deposited | Quartet | 500 |
| WAAM 2 | Figure 12b | Ti-6Al-4V | As deposited | Quartet | 500 |
| L-BPF 3 | Figure 12c | Ti-6Al-4V | As deposited | Quartet | 500 |

### 2.3.1. Roughness Comparators

Table 2 lists the roughness comparators used in this study, along with their minimum and maximum $R_a$ values. The cast comparator was originally specified in microinch RMS and has been converted to $R_a$. Comparators were manufactured by Rubert + Co., Ltd. (ACRU Works, Cheadle Hulme, Cheadle, UK), and GAR Electroforming (Danbury, CT, USA). Examples of two comparators, shot blast and spark erosion, are shown in Figure 3.

**Table 2.** Overview of roughness comparators.

| Finish Type | Lowest $R_a$ μm | Upper $R_a$ μm |
|---|---|---|
| Honing | 0.05 | 1.6 |
| Shot blasting | 0.4 | 12.5 |
| Spark erosion | 0.4 | 25 |
| Grit blasting | 0.4 | 12.5 |
| Turning | 0.4 | 12.5 |
| Horizontal milling | 0.4 | 12.5 |
| Vertical milling | 0.4 | 12.5 |
| Grinding | 0.05 | 1.6 |
| Lapping | 0.05 | 1.6 |
| Cast | 0.5 | 20.5 |

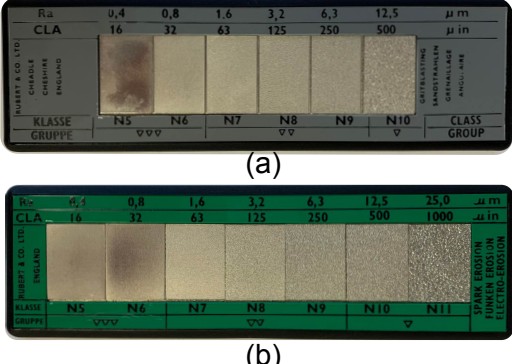

**Figure 3.** Photographs of two roughness comparators, (**a**) shot blast and (**b**) spark erosion.

The comparators are made by nickel electroforming and are assumed to have a grain size much smaller than the resolution of the SRAS instrument and free from significant

orientation texturing, such that the microstructure may be considered homogeneous across each comparator.

Small $4 \times 4$ mm regions were scanned using the Quartet detector on the grit blast sample with 100 μm step size. To obtain the data on SNR and dropout, all of the comparators were scanned with 10 lines and a step size of 100 μm across the full width of the surface area, covering $1 \times 10$ mm to $1 \times 20$ mm depending on the width of each comparator. For each point in the scan area, the returning light level and velocity were calculated. The SNR was calculated by comparing the signal height in the Fourier domain to the average height of a noise trace taken at the same time and expressed in decibels. The number of lost data points (dropout) varies with surface roughness and was calculated by setting a threshold on the optical return of 0.1 (arbitrary units), which excludes data where insufficient light returns to the detector to give usable data.

For the grit blast sample, this process was repeated with a different number of averages, so the change in SNR and velocity standard deviation with averages could also be determined.

### 2.3.2. Etched Surfaces

A Ti-6Al-2Sn-4Zr-2Mo-Si specimen ($82 \times 32 \times 2$ mm) was prepared by standard metallographic preparation for titanium and etched with Kroll's reagent to reveal the grain structure through optical macrography. The Quartet detector was used to scan the sample, with 16 averages per point and a pixel size of $100 \times 200$ μm.

### 2.3.3. Machined Components

Two samples with machined surfaces have been examined. The first was a large-grained commercially pure titanium sample, which was first prepared and polished for SRAS scanning. Two subsequent milling operations were then performed on the top surface, giving an $R_a$ of 1.6 μm and 3.2 μm, respectively. The sample was re-scanned after each milling procedure. The scans of $25 \times 23$ mm of the sample were performed with the Quartet detector using 1000 averages per point and a $100 \times 100$ μm step size.

A second sample of a Ti-6Al-4V disk (260 mm diameter, 13 mm thickness), with a surface roughness of $\sim 0.8$ μm from milling was also scanned. The SKED detector was used to scan the whole surface area with 16 averages per point. The whole disk was scanned with a fixed angular step; it leads to the step size of $100 \times 200$ μm–$100 \times 400$ μm in the top half from the inner ring to the outer ring and $100 \times 2$ μm–$100 \times 2180$ μm from the centre of the disk to the outer ring in the bottom half.

### 2.3.4. Additive Manufacturing Specimens

This article includes results from three specimens produced using the WAAM process with Ti–6Al–4V welding wire feedstock onto a titanium substrate. A pulsed gas tungsten arc welding torch, with argon shielding, was used to deposit structures made of 20 single-bead layers, giving an approximate final width of ~6 mm and height of ~24 mm. Following the deposition and cooling of each layer, a 100 mm diameter roller was passed over the top surface to create the deformed specimens. The rolling applies vertical forces of 50 kN or 75 kN, which is monitored by a load-cell. More information on the build process used to produce these specimens can be found in [40], whilst further detailed information on the WAAM technique can be found in Martina et al. [41]. The Quartet detector was used to scan a $8 \times 12$ mm area using 2000 averages per point with a step size of $100 \times 100$ μm. A final WAAM specimen was made from a single deposition bead of Ti-6Al-4V. An area of $36 \times 16$ mm was scanned using 500 averages and a step size of $100 \times 100$ μm.

A further three specimens manufactured by L-PBF are also presented in this work. Two (Al-block and Ti-cube) were manufactured using a Realizer SLM50 (ReaLizer GmbH, Borchen, Germany) equipped with a 100 W continuous wave laser at 1064 nm wavelength. The laser was used to melt and fuse powder feedstock layer-by-layer to build the sample.

In the case of the Al-block, AlSi10Mg powder was used, and Ti-6Al-4V powder was used for the Ti-cube.

The titanium sample was scanned with the SKED detector over an area of $10 \times 10\,\mathrm{mm}$, using 128 averages per point and a step size of $50 \times 50\,\mathrm{\mu m}$.

The Al-block was designed with six side holes, with diameters ranging from $0.5\,\mathrm{mm}$ and $1\,\mathrm{mm}$. The sample was scanned with the Quartet detector, and the top face used 200 averages and a $17 \times 9\,\mathrm{mm}$ scan area with a step size of $100 \times 200\,\mathrm{\mu m}$. A $7 \times 5\,\mathrm{mm}$ area on the side wall was also measured, using 8000 averages and a step size of $100 \times 67\,\mathrm{\mu m}$. The side wall was particularly challenging to work on due to the extreme level of roughness caused by unmelted particles sticking to this surface and so required more averages to obtain a good SNR than the top face. Further detail on the Al-block and Ti-cube can be found in Pieris et al. [42] and Patel et al. [43], respectively.

The third L-PBF sample, an SRAS-sample, was manufactured in Ti-6Al-4V using a Renishaw AM250 (Renishaw plc., Wotton-under-Edge, UK). This was measured with the Quartet detector over an area of X by Y mm, using 4000 averages per point and a step size of $100 \times 100\,\mathrm{\mu m}$.

A final WLAM sample was fabricated from Ti-6Al-4V, with the addition of boron towards the build centre. The deposition was built in a circular pattern on a titanium substrate. This was scanned with the Quartet detector over an area of X by Y mm using 2000 averages per point and a step size of $100 \times 100\,\mathrm{\mu m}$.

## 3. Results

The results presented start with the standardised roughness comparators measured with the same instrument and detector; they explore the impact that different surface preparations have on the ability to perform measurements. These are followed by measurements on a variety of sample surfaces that exhibit different degrees of form, waviness, and roughness.

### 3.1. Roughness Comparators

The effects of a rough surface on the optical generation and detection of surface acoustic waves can first be demonstrated by examining the results from SRAS measurements from the shot-blast comparator. Figure 4a demonstrates the simple effect of increasing roughness resulting in less light being returned to the detector. It is seen that the average light returned from the sample decreases with increasing roughness up to $R_a$ up to $3.2\,\mathrm{\mu m}$ and then stabilises around 15% for increasing levels of roughness. The data dropout represents the number of unmeasurable pixels across the surface and is seen to increase with increasing roughness, unlike the light return level, which does not plateau. For this surface type with increasing roughness, there is a decreasing number of positions on the surface that will produce a measurement; however, the level of light returned from the measurable positions will remain roughly constant.

Figure 4b plots the standard deviation of the velocity for pixels where the light return is over a chosen threshold against the number of averages used. The first conclusion to draw from this figure is that for a given number of averages, the standard deviation of the velocity increases with increasing roughness. As the number of averages increases, the standard deviation of the velocity decreases; however, it does not drop as quickly as would be expected if the noise was uncorrelated. A 100-fold increase in averages should produce a 10-fold reduction in the standard deviation for random noise, but the drop is around 2–3 times. There are a number of possible reasons for this; for the rougher samples there are two effects, an increase in the acoustic attenuation, leading to a reduction in signal level at the detection location, and secondly, the roughness causes a drop in the return light level, leading to a higher level of noise in the detection channel. Additionally, the data come from different spatial locations on the sample, and although we believe the grain size is very small compared to the measurement resolution, variations in the material could also lead to a plateau in the standard deviation of the velocity with averages. Another source of noise

is the Q-switch in the generation laser being picked up by the detection electronics, and as this repeats with each generation event. This noise is (partially) coherent and so does not average away as quickly as uncorrelated noise. For the samples where the light return is lower, the effect of the background noise is more marked, and even with a large number of averages, the velocity standard deviation is still quite high. However, reasonable results are still obtained with velocity standard deviation below $\sim 50~\mathrm{ms}^{-1}$, achievable for all samples; this represents an error on the mean of $\sim 2700~\mathrm{ms}^{-1}$ of below 2%.

Figure 4c plots the mean velocity for $R_a$ between 0.4 µm and 6.3 µm; the velocity is seen to decrease from $\sim 2715$ to $\sim 2630~\mathrm{ms}^{-1}$. There are two possible hypotheses that may explain this change in wave speed. The first may be related to the defocus in the generation patch, meaning the wavelength of the acoustic wave generated is not consistent. The second possibility is a dispersion of the SAW due to the roughness. The second effect seems more plausible, as for the roughest samples, the focus position was checked. Furthermore, to obtain a change of $\sim 85~\mathrm{ms}^{-1}$ from defocus would require a 10% change in acoustic wavelength. However, for the case of dispersion due to surface roughness, this is expected to only reduce the velocity of the wave and so is the likely cause of the small shift in mean velocity observed.

Finally, Figure 4d summarises these effects by overlaying SRAS velocity maps from each surface onto a photograph of the comparator. The overall quality of the measurement decreases with increasing roughness as more dropouts are seen, and the shift in mean velocity is apparent for the roughest samples.

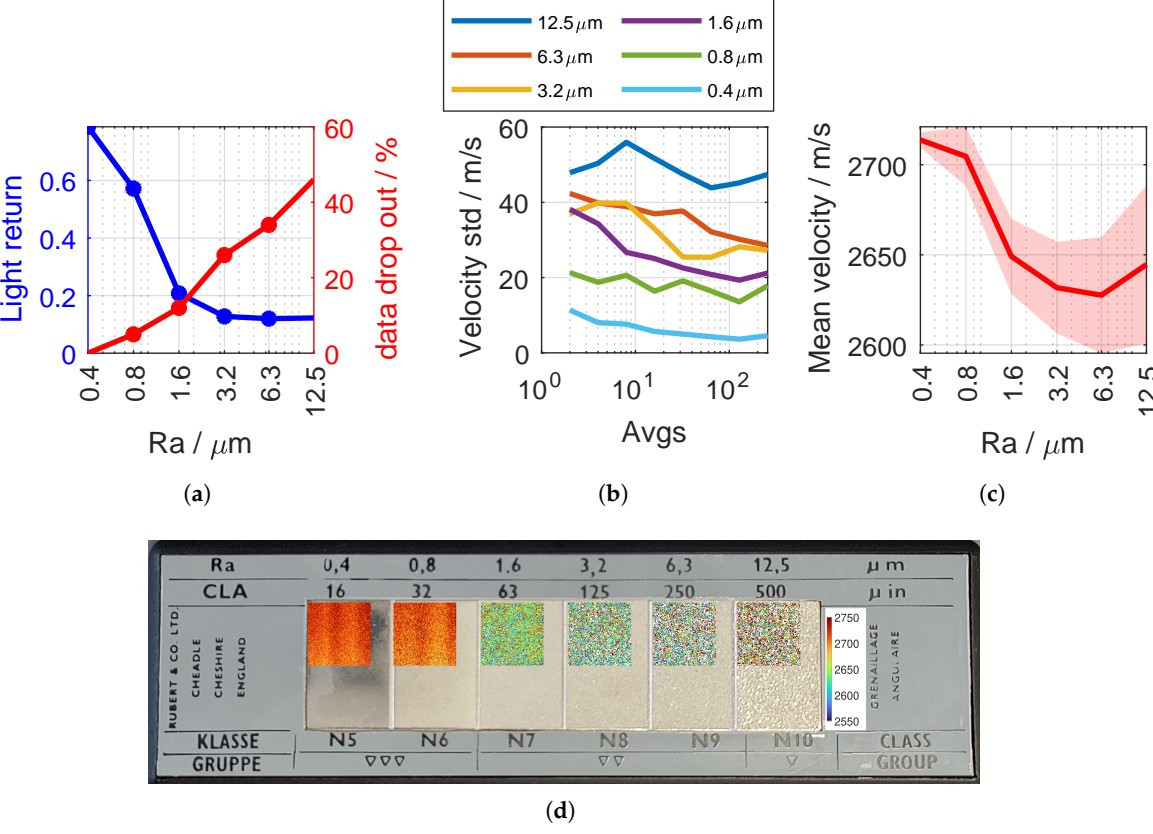

**Figure 4.** Results from grit blast comparator demonstrating the key features of laser ultrasound measurements on rough surfaces. (**a**) The light returned to the detector (blue) and percentage of positions on the sample where no measurement could be made (red). (**b**) Velocity standard deviation with averaging. (**c**) Mean velocity (bold) with increasing roughness, where the filled region represents velocity standard deviation. (**d**) SRAS velocity maps overlaid onto the grit blast comparator from which they were captured.

It would be desirable to have a direct relationship between surface roughness and the quality of the ultrasonic measurement. Figure 5 shows that the roughness does not correlate with the SNR obtained. It is easy to demonstrate why this is the case by examining the reflected beams from surfaces of the same notional roughness. Figure 6a–d show an image of the reflected beam from four samples that all have an $R_a$ of 0.8 μm. It can clearly be seen that machining methods in particular can impart a strong directionality onto the reflection.

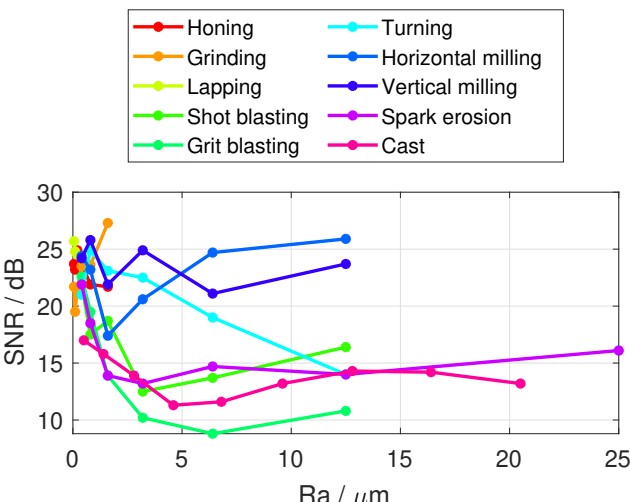

**Figure 5.** Surface roughness versus SNR; the $R_a$ does not generally correlate with the SNR obtained.

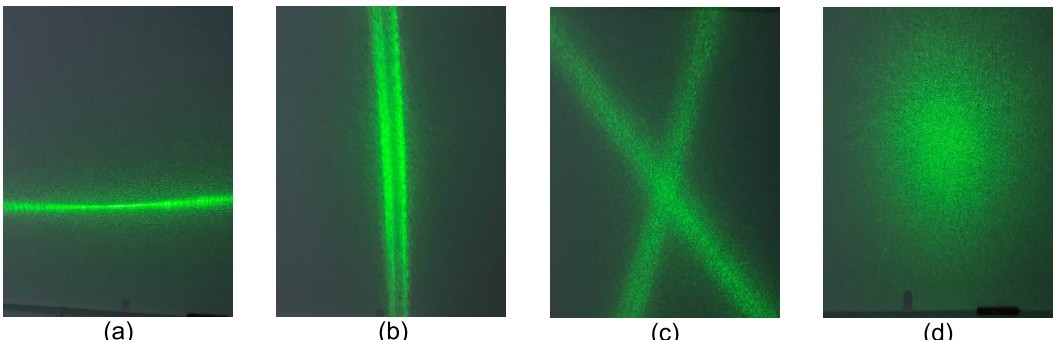

**Figure 6.** Comparison of reflections from four specimens with a surface roughness of 0.8 μm. Despite having the same $R_a$ value, there is a significant difference between the reflections. In (**a**) horizontal milling, (**b**) vertical milling, and (**c**) honing, highly directional responses are seen, with the light following the direction of the machining tool. In contrast, a nearly Gaussian reflection is seen from (**d**) grit blasting.

Tabulated results for the comparators are given in Tables 3 and 4, reporting the average single-shot SNR when there is sufficient light returned from a point on the surface and the percentage of lost data points for each sample. These tables present the difficulty of working on different surface types and degrees of roughness. The SNR table explains how good data are likely to be from this surface if sufficient light is returned, and the dropout table explains how good an image of the surface is likely to be.

In general, if light is returned to the detector, then the single-shot SNR is good from ~11 to 27 dB. For most surfaces, the SNR is similar across the range of roughness levels present. In some cases, the roughest surface actually produces higher SNR data, for example, in grinding, which is likely due to the increase in the correlation length for the roughest sample; the feature sizes look significantly coarser to the eye, reducing the impact of the roughness on the ultrasound propagation.

Two samples show a marked drop in SNR with increasing roughness (grit blasting and spark erosion). These surface preparations show that as the roughness increases, as does the feature size (or correlation length); this is essentially starting as a surface where small-scale roughness dominates and then becomes a surface where waviness dominates, eventually producing features with such large scale they can be considered the form. Roughness leads to some scattering of the light, which, coupled with large surface angles for wavy features, leads to even more light being reflected outside of the collection optics, leading to a significant drop in light return.

For two samples, honing and turning, the surfaces have very pronounced curvature, so there is a large change in surface height across the sample. This leads to a large number of lost points when the surface goes out of focus. When the surface is in focus, the data have generally good SNR. These types of surfaces would benefit from being able to ensure the sample is at the focal plane and normal to the surface. This requirement is discussed further in Section 4.1.

**Table 3.** Single-shot average SNR (dB) for all valid data points taken on each roughness comparator using the Quartet detector.

| Finish Type | Single Shot SNR (dB) | | | | | | | | | |
|---|---|---|---|---|---|---|---|---|---|---|
| $R_a$ / µm | 0.05 | 0.1 | 0.2 | 0.4 | 0.8 | 1.6 | 3.2 | 6.4 | 12.5 | 25 |
| Honing | 23.7 | 23.2 | 24.9 | 23.7 | 21.9 | 21.7 | ◇ | ◇ | ◇ | ◇ |
| Grinding | 21.7 | 19.5 | 21.4 | 23.5 | 23.6 | 27.3 | ◇ | ◇ | ◇ | ◇ |
| Lapping | 25.7 | 24.8 | 24.5 | ◇ | ◇ | ◇ | ◇ | ◇ | ◇ | ◇ |
| Shot blasting | ◇ | ◇ | ◇ | 21.9 | 17.5 | 18.7 | 12.5 | 13.7 | 16.4 | ◇ |
| Grit blasting | ◇ | ◇ | ◇ | 22.6 | 19.5 | 13.9 | 10.2 | 8.8 | 10.8 | ◇ |
| Turning | ◇ | ◇ | ◇ | 21.0 | 24.9 | 23.1 | 22.5 | 19.0 | 14.0 | ◇ |
| Horizontal milling | ◇ | ◇ | ◇ | 24.4 | 23.2 | 17.4 | 20.6 | 24.7 | 25.9 | ◇ |
| Vertical milling | ◇ | ◇ | ◇ | 24.2 | 25.8 | 21.9 | 24.9 | 21.1 | 23.7 | ◇ |
| Spark erosion | ◇ | ◇ | ◇ | 21.9 | 18.5 | 13.9 | 13.2 | 14.7 | 14.0 | 16.1 |
| $R_a$ / µm | 0.5 | 1.4 | 2.8 | 4.6 | 6.8 | 9.6 | 12.8 | 16.4 | 20.5 | |
| Cast | 17.0 | 15.8 | 13.9 | 11.3 | 11.6 | 13.2 | 14.3 | 14.2 | 13.2 | |

◇—surface of this roughness and finish type not available for testing.

**Table 4.** Drop out for each roughness comparator, i.e., ratio of unmeasurable positions to measurable positions on the sample surface.

| Finish Type | % Drop-Out | | | | | | | | | |
|---|---|---|---|---|---|---|---|---|---|---|
| $R_a$ / µm | 0.05 | 0.1 | 0.2 | 0.4 | 0.8 | 1.6 | 3.2 | 6.4 | 12.5 | 25 |
| Honing | 35 | 41 | 38 | 35 | 41 | 45 | ◇ | ◇ | ◇ | ◇ |
| Grinding | 0 | 1 | 1 | 0 | 1 | 2 | ◇ | ◇ | ◇ | ◇ |
| Lapping | 0 | 1 | 1 | ◇ | ◇ | ◇ | ◇ | ◇ | ◇ | ◇ |
| Shot blasting | ◇ | ◇ | ◇ | 0 | 5 | 5 | 17 | 13 | 23 | ◇ |
| Grit blasting | ◇ | ◇ | ◇ | 0 | 5 | 10 | 16 | 26 | 41 | ◇ |
| Turning | ◇ | ◇ | ◇ | 57 | 21 | 21 | 27 | 45 | 59 | ◇ |
| Horizontal milling | ◇ | ◇ | ◇ | 1 | 2 | 20 | 6 | 3 | 3 | ◇ |
| Vertical milling | ◇ | ◇ | ◇ | 1 | 1 | 0 | 2 | 13 | 4 | ◇ |
| Spark erosion | ◇ | ◇ | ◇ | 0 | 1 | 2 | 2 | 13 | 30 | 38 |
| $R_a$ / µm | 0.5 | 1.4 | 2.8 | 4.6 | 6.8 | 9.6 | 12.8 | 16.4 | 20.5 | |
| Cast | 10 | 20 | 44 | 55 | 42 | 41 | 36 | 51 | 75 | |

◇—surface of this roughness and finish type not available for testing.

### 3.2. Wavy Samples

WAAM samples have varying degrees of *form*, with high levels of *waviness* but less overall fine-scale *roughness*. Figure 7 shows the optical micrographs from three WAAM samples with different processing parameters. The top sample is built layer-by-layer without a rolling pass. The middle and bottom samples have an interpass rolling stage with 50 kN and 75 kN force applied, respectively. The optical images clearly show the layer-by-layer nature of the build and for the undeformed sample, additional significant

colouration of the surface due to surface oxide/nitride layers forming during the time spent at elevated temperatures.

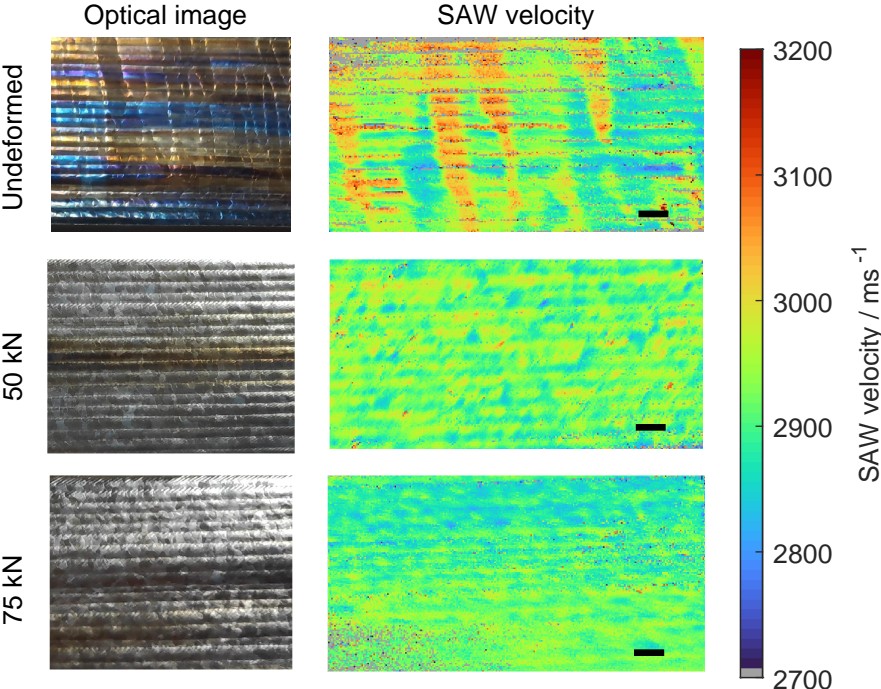

**Figure 7.** Optical photographs (**left column**) and SRAS velocity maps (**right column**) from sidewall of as-deposited Ti-6Al-4V WAAM specimens in (**top row**) sm undeformed state and post- (**middle row**) 50 kN and (**bottom row**) 75 kN interpass rolling. Scale bar in velocity maps indicates 2 mm.

The velocity maps show the underlying texture of the samples; for the undeformed case, there is the presence of large prior-β grains, i.e., the long columnar growth that continues up through the layers as the sample is built. The interpass rolling stage prevents this from happening, where the influence of this grain refinement step breaks up the grains and produce a finer texture. These scans on the as-built surface show the same information as previous work on polished specimens [40].

There are some drop outs in the data, caused by the change in angle of the surface at the bead joins, the wavy nature of the side wall means the light is sometimes reflected out of the instrument. The local roughness is not influencing the measurement as the surface on the small scale is quite smooth. The discolouration of the undeformed sample had little impact on the determination of the velocity as it only reduced the return light level of the probe laser by less than 20%.

### 3.3. Rough Samples

The etched sample has no *form* or *waviness*, just fine-scale *roughness* from the etching process. Figure 8 shows a velocity map captured from the etched Ti-6246 specimen overlaid on to a photograph of the surface, with the grain structure visible thanks to etching. The etching process produces roughness on a fine scale with different materials and processes producing different feature sizes scattering the light to different degrees. For this sample, despite the strong contrast in reflectance and the roughness imparted on the surface from the etching process, the SRAS measurement is still possible, with minimal data dropout of 4% and clear contrast between the grains in the SAW velocity map.

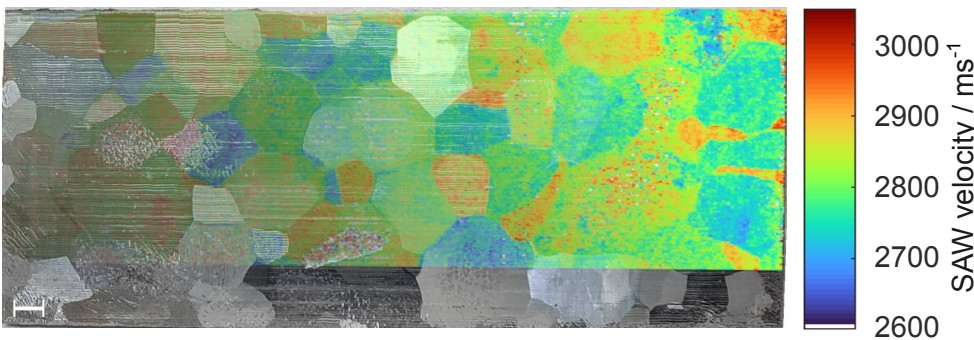

**Figure 8.** Optical micrograph of an etched Ti-6246 sample with overlay of velocity map, taken with 16 averages per point. The scale bar shown in the lower left-hand corner indicates 5 mm. The transparency of the velocity map is used to allow the etched grain structure to be shown for comparison.

L-PBF samples were designed to have no *form* or *waviness* so that the small-scale *roughness* could be examined in isolation. L-PBF currently produces very challenging surfaces for laser ultrasound inspection, as the surface roughness is related to the particle size used for the powder layer. For the samples used in this study, this is around 30 μm and leads to an $R_a$ in the range of 10 μm–15 μm, with a similar correlation length. Given that the acoustic wavelength is around 80 μm and the focal spot for the probe is around 30 μm, this has a large impact on both the generation and detection of ultrasound.

Figure 9a shows two samples made via L-PBF. The left shows a top surface scan on a Ti-6Al-4V cube scanned using the SKED (see Figure 2b). Here, there are a considerable number of dropouts due to the extremely rough surface. The velocity measured shows little contrast due to the small grain size and strong texturing, in line with previous results in polished specimens [44]. By changing the detection spot distance from the generation patch, it is possible to combine scans to reduce the drop-out rate. The combination of six scans reduces the dropout from around 50% to below 10% at the cost of total scan time.

Figure 9b shows results using the Quartet detector on an AlSi10Mg sample made with internal defects [42], the velocity map of the top, and a small section of the side wall around the side hole feature. Again, there are some drop outs, but the texture information is visible in the velocity maps. The two velocity maps are plotted on the same velocity scale, indicating a difference in the dominant crystallographic plane between the two faces.

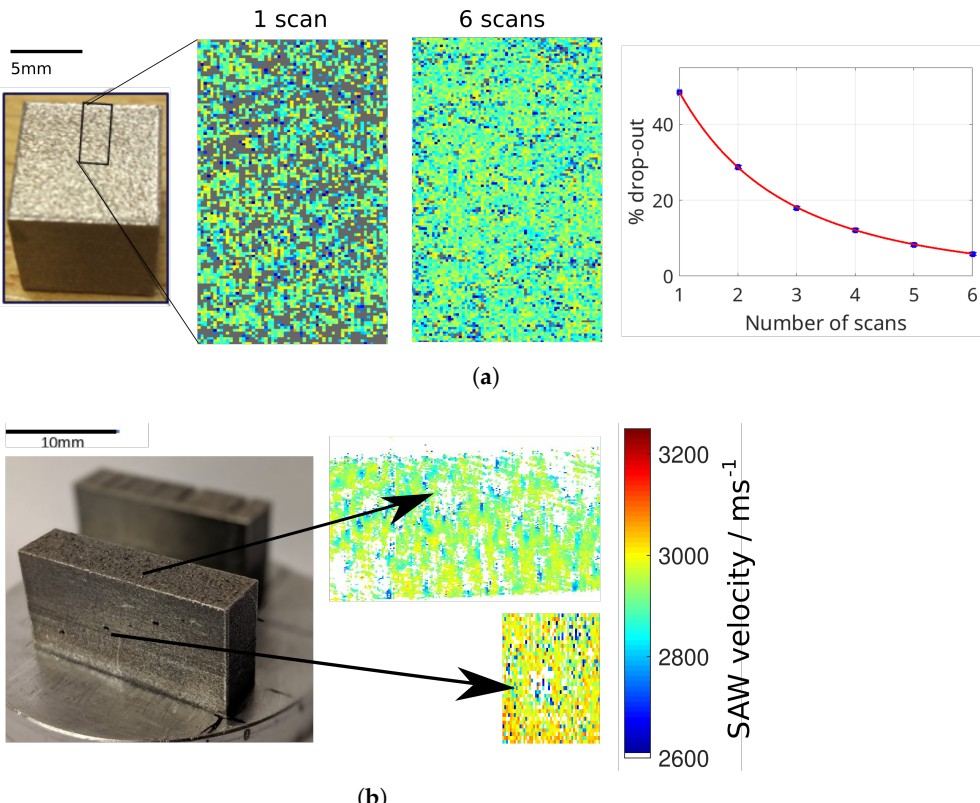

(a)

(b)

**Figure 9.** LPBF: (**a**) Ti-6Al-4V, acquired using the SKED showing that combining scans with different detector positions can reduce data loss. (**b**) Al sample with designed defects measured on the top and side wall with the Quartet detector.

### 3.4. Wavy and Rough

The machined samples presented here are predominantly flat with little *form*, but depending on the machining depth, they can have *waviness* along with fine-scale *roughness*.

A commercially pure titanium sample with three surface conditions, polished (top row) and milled to $R_a$s of 1.6 and 3.2 µm, has been measured. The left column of Figure 10 shows the optical maps from each surface, with the right-hand side presenting the corresponding velocity map. The rough surface maps are very similar to that in the polished state. Variations between the pictures are due to several factors; for example, removal of material from the polished state exposes slightly different grains, which is particularly noticeable where the features are small. Additionally, small variations in the sample placement in the instrument lead to small rotational differences that influence the velocity to different degrees depending on the plane of each grain. A small change in the rotation of the sample for the large blue grain, which appears to be aligned with the prism plane, can cause a larger shift in velocity than for other grains.

A machined titanium (Ti-6Al-4V) billet slice ($R_a \approx 0.8$ µm) was scanned. A selection of the optical image is overlaid on the velocity map in Figure 11. This sample was scanned by rotating the billet and scanning a ring of data before moving the SRAS head sideways across the centre line of the sample. This was to produce data sets to mimic in silico scanning in a commercial lathe. The wave propagation direction changes for each position on the sample, and in this case, the waves are propagating towards the centre of the sample. Despite the surface roughness, 90% of the surface produces valid SRAS measurements, and the machining lines are not shown in the velocity maps. The structure within the velocity maps shows the fine microstructural texture of the material.

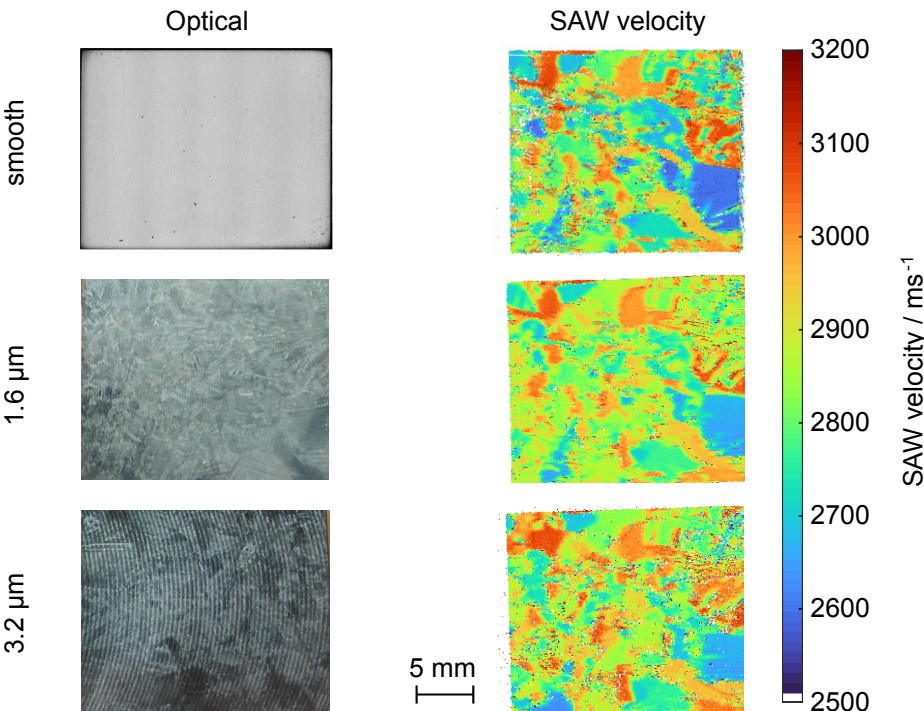

**Figure 10.** Commercially pure titanium sample; the sample was polished, then milled twice over, creating a rough surface, with SRAS measurements captured after each step. Optical maps of the surface are shown in the left column with velocity maps to the right.

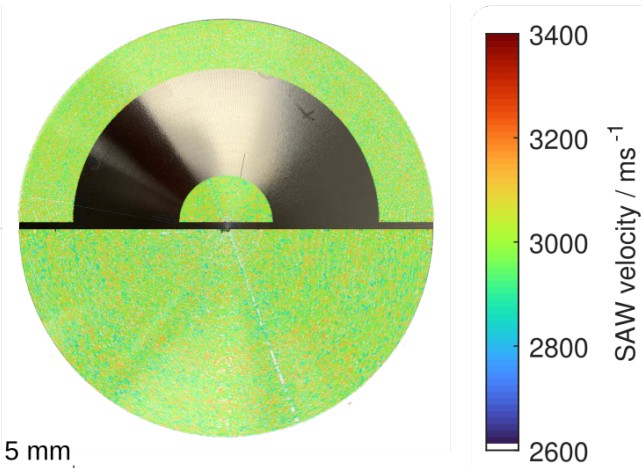

**Figure 11.** SRAS velocity map of machined Ti-6Al-4V billet, with surface roughness of $R_a \approx 0.8\,\mu m$. The lower half of the billet shows a velocity map captured at a fixed angular resolution, leading to an increasing pixel size towards the edge of the sample. In contrast, the velocity maps in the upper half maintain a constant pixel size. A cropped photograph of the sample is inset in the upper half to show the surface finish.

The lower half of the billet reports the results of scanning radially. Maintaining a constant angle between scan lines, this means that the pixel size increases from the centre towards the outer edge. In contrast, the upper half of the scan is such that constant pixel size is maintained across the scan area. Therefore, the outer and inner rings are consistent. Careful choice of scan strategy is required to obtain well-sampled data.

## 4. Discussion and Conclusions

The following section discusses the challenges of using SRAS for the inspection of real parts and considers the source of the challenges from samples with waviness and roughness and possible solutions. The time to inspect compared to the build time is also considered, as well as other effects encountered when inspecting close to the build in AM, such as elevated temperature. Finally, prospects for the future are given.

### 4.1. Challenges Due to Waviness

The previous section showed the capability of SRAS measurements on a range of surface finishes. It is clear that for many real parts, the fact that these surfaces represent small-scale roughness is not an impediment to microstructural imaging.

The current versions of the instrument suffer mostly from challenges from the form and waviness of the samples. This is because, for large changes in the sample surface position (as the form changes) the measurement surface goes out of focus, or the light reflects and leaves the optical system due to the change in surface gradient. Some examples of this are shown in Figure 12.

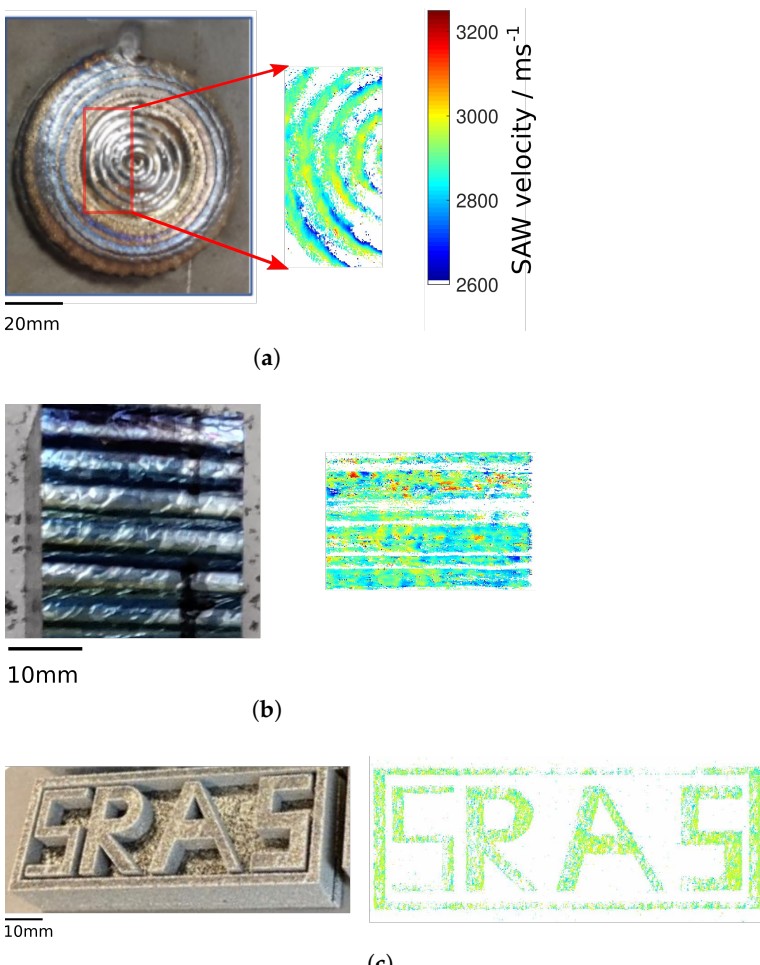

**Figure 12.** (**a**) Titanium diboride (TiB2) made by WLAM, where the build has large changes in surface height across the surface. (**b**) Ti 64 made by WAAM, where the build strategy incorporates single-weld beads producing deep ridges between layers. (**c**) Ti 64 sample made by L-PBF, with both large changes in height between features and fine-scale roughness from the powder.

The influence of changes in focus could be reduced by incorporating an auto-focus system, either via image analysis of the surface and the generation fringe pattern, or by sensing the stand-off distance variation from the sample via other means. Adding this to

the instrument will allow the measurement surface to be kept at the optimal distance and so greatly improve signal integrity.

Maintaining the measurement plane normal to the surface to combat changes in form and waviness is more challenging. For parts where there is a CAD model, registering the part in the measurement volume and using a robotic manipulator to present the surface normal to the laser head would allow the impact of changes in surface gradient to be reduced. Combining this with auto-focus would produce a very robust measurement system.

There are still challenges from wire-based AM samples where the side wall of the sample is wavy and the exact form of this is unknown and therefore hard to follow. In Figure 12a, the waviness of the WLAM sample shows clear drop outs, but the data from the upper flat regions are good. Similarly, for the WAAM sample in Figure 12b, the variation in layer thickness and bead curvature would be difficult to follow without a dedicated surface following system built into the instrument. Again, where the curvature is small and the sample is in focus, the data obtained are still good as the roughness is well within the operating range of the instrument.

### 4.2. Challenges Due Roughness

Of the materials presented, L-PBF presents the biggest challenge from the small-scale roughness perspective (Figure 12c). However, it is important to note that the optimisation of the L-PBF process to reduce surface roughness is an ongoing research domain, and within the last few years clear progress has been made. Therefore, it is realistic to foresee a future where the ever-increasing capability of detector systems and smoother as-built surfaces of L-PBF mean that measurements on unprepared parts become routine.

In addition, the surface roughness of L-PBF components are lower than any other powder based-technique [45]. The results presented in this paper deal only with L-PBF and not other varieties of powder-bed manufacture, such as selective laser sintering (SLS), but it is reasonable to assume the roughness seen in SLS components would prove a greater challenge to SRAS measurements.

The dropouts are primarily caused by loss of detection light and so multiple scans can be combined and stacked to reduce this effect as discussed earlier. Another approach is to broaden the detection laser spot into a line or a number of spots spaced wider than the correlation length of the roughness so that some light will always return to the sample for each scan location.

### 4.3. Resolution

As previously detailed by Smith et al. [16], the spatial resolution of an SRAS measurement is approximately equal to half the diameter of the generation patch. Therefore, to increase spatial resolution, two approaches are available. First, the number of fringes can be reduced; however, this results in a broadband signal and a greater error in velocity measurement. The second approach is to use a smaller acoustic wavelength, giving a smaller patch size for the same number of fringes, but this introduces two separate challenges. Using smaller wavelengths requires the detection of higher frequencies, and so the bandwidth of the detector used in the measurement can become a limiting factor. As shown in Figure 2, whilst prepared surfaces can be measured in the hundreds of MHz by the KED, and indeed other solutions exist to measure SAWs in the GHz regime [46], rough surfaces are limited to 100 MHz, with many detectors having an upper limit below 50 MHz. Additionally, the attenuation of the SAW as it propagates must also be considered. The short propagation distances in SRAS help to minimise the influence of attenuation, and increasing the frequency to improve resolution will eventually lead to the attenuation dominating. As derived and observed experimentally by Huang and Maradudin [22], the attenuation of a SAW due to roughness varies with the frequency to the fourth power in the stochastic regime; further detail on the can be found in the recent work of Sarris et al. [47].

The results presented in this work have an approximate spatial resolution of 400 μm; the inherent flexibility of the SRAS instruments makes it possible to switch between op-

timising for spatial resolution and optimising for velocity resolution. The need to work at lower frequencies usually means spatial resolution is often sacrificed for samples with rough surfaces. The velocity resolution is determined by the number of fringes used in the generation pattern and the SNR achieved via averaging, and so a trade-off between spatial resolution and scan time is usually made.

### 4.4. Scan Speed Compared to Manufacturing Speed

An obvious question to ask is what the speed of an SRAS scan is. Whilst it is plain that a 'fast' scan is more amenable to integration within a manufacturing process, some context can be provided by using L-PBF as worked example. This allows the speed of an SRAS scan to be compared to the speed of the AM deposition. Whilst the example herein focuses on L-PBF, a similar methodology could be applied to other methods of additive manufacturing or machining, so long as the processing rates of the manufacturing operation are known.

Colosimo et al. developed a model for assessing the economic impact of in silico monitoring tools in AM [3]. The authors conducted a series of studies to estimate the failure rate of typical AM components. They then calculated the break-even point of a hypothetical inspection system as a function of this failure rate. The conclusion was an estimation that ~9% savings in the cost of additive components could be achieved by the use of in silico monitoring. All of the tools the authors considered were passive, observing the emission spectra of the build process. These are essentially 'free' measurements in that there is no temporal impact on the build; therefore, only the cost of the equipment needs to be considered. However, a prospective SRAS system will need to scan the sample, causing some addition to the manufacturing time, hence the importance of the time element of the model compared to those in the existing literature. To generate the build time information shown below, a sample build similar to the Airbus A320 nacelle hinge bracket discussed in the work of Tomlin and Meyer [48] was defined. This bracket was later used by Colosimo et al. as a case study example in the aforementioned study [3]. The part takes approximately 36 h to fabricate.

To estimate the time spent inspecting the sample build, the 'speed' of the inspection system must be defined. The 'speed' of the SRAS system can be essentially broken down into the rate of data acquisition and the number of data points to be captured. The speed of acquisition is a function of the number of averages required and the repetition rate of the generation laser. The number of points to be captured is primarily dependent on the size of the specimen and the acoustic wavelength (as this in turn defines the spatial resolution). Figure 13 plots the impact on build time as a function of points acquired per second and the wavelength of inspection. To give some context, parameters from the results presented earlier in this paper are marked on this figure.

The closer to one the ratio is, the smaller the impact of inspection on the total build time. On first inspection, it would seem desirable to work as close to the lower right-hand side as possible in order to minimise the temporal penalty of inspection; however, increasing the wavelength of inspection leads to a decreased resolution and decreased measurement fidelity. The discontinuities arise because with increasing wavelength, more build layers can be inspected per scan. Therefore, increasing the acquisition rate of the SRAS system is the preferable way to increase scan speeds. The number of parameter points per second shown in Figure 13 is simply $\frac{\text{rep rate}}{\text{averages}}$.

On a smooth surface, the primary influence on the points per second is the repetition rate of the generation laser. In the 'on-the-fly' scanning approach [16], as used by the smooth surface system, one pulse is equivalent to one measurement point—thus, a 5 kHz repetition rate laser allows 5000 measurements per second. However, on rough surfaces, the signal-to-noise ratio (SNR) is significantly lower and necessitates averaging of the acoustic waveform to improve SNR.

It would be preferable to correlate the number of averages needed with surface roughness and integrate this within the model; however, as previously discussed, this relationship is extremely complex.

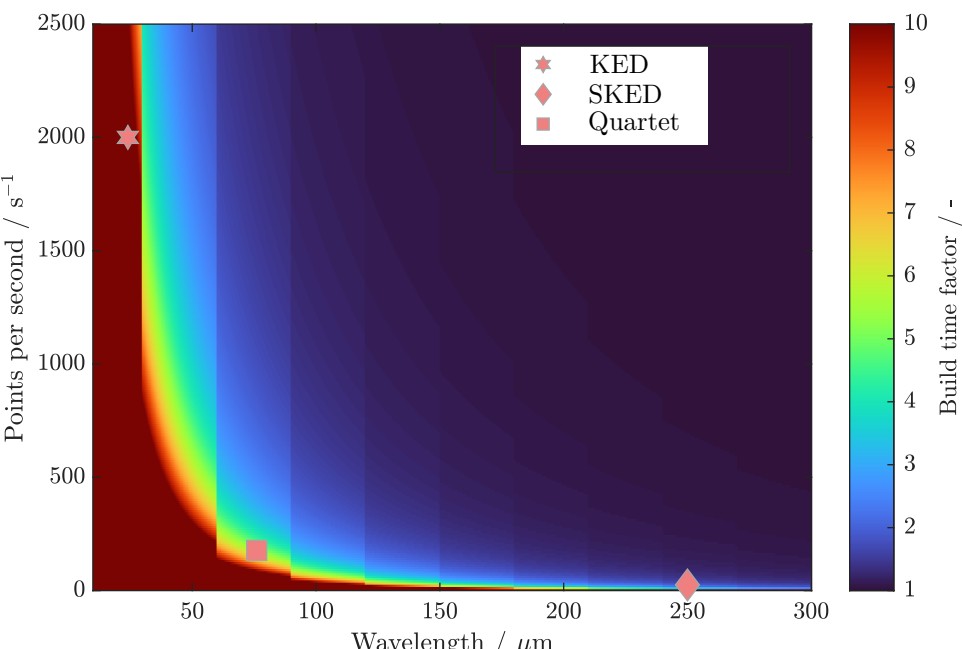

**Figure 13.** Ratio of build + scan time to build time, as a function of inspection wavelength and data points captured per second. Marked points show the parameter for three of the detectors, where KED is single-shot and high-resolution but requires smooth surfaces, and SKED was on L-PBF sample with 116 averages with 250 μm wavelength, and Quartet was from the etched sample, using 16 averages and 80 μm wavelength.

### 4.5. Temperature

As discussed, the ability to make SRAS measurements on rough surfaces raises the possibility of integrating SRAS within the manufacturing process, which increases the final challenge discussed in this work, i.e., temperature. The acoustic velocity of material is sensitive to the temperature, and thus the use of laser ultrasound in steel mills, L-PBF chambers, and direct energy deposition (DED) systems requires calibration of the temperature. By way of example, the SAW velocity on the (001) plane of nickel at temperatures between 100 and 760 K displays a near-linear trend of $-0.4\,\mathrm{ms^{-1}\,K^{-1}}$. Thus, a temperature instability of a few degrees can become the limiting factor in the accuracy velocity measurements and orientation determination.

Experimental results presented in this work have been at nominal room temperature, but this was not controlled and probably spans a range of between 5 and 10 °C. To work at higher temperatures would require accurately measuring the temperature during a scan so that the temperature effect can be calibrated.

### 4.6. Future Perspectives

The images presented here show that real surface conditions of parts are no longer an impediment to imaging the microstructure. Future generations of SRAS instruments will be able to deal with the complex shapes of parts, allowing the surface microstructure to be determined on the part before being put into service. This will allow decisions on inspection and maintenance schedules at the part level, reducing costs and time associated with inspection. It will also accelerate the adoption of additively manufactured components into safety-critical areas, as the performance of the part will be known because the part can be inspected during the build process, generating a digital twin of the part containing information on the remaining defects and microstructure within the part. The availability of optical detectors that can work on a wide range of surface finishes means that imaging the microstructure of real parts is now possible.

### 5. Patents and Commercialisation

Both the SRAS technique and SKED technology are covered by patents WO2007003952A3 and WO2013079960A1, respectively.

**Author Contributions:** Conceptualization, R.J.S. and M.C.; methodology, R.J.S., W.L. and P.D.; Experiments, W.L., P.D. and R.P.; writing—original draft preparation, W.L., P.D., R.P., D.P. and R.J.S.; writing—review and editing, all. All authors have read and agreed to the published version of the manuscript.

**Funding:** This work was supported by the Engineering and Physical Sciences Research Council [Grant No. EP/S013385/1].

**Data Availability Statement:** The data presented in this study are available on request from the corresponding author. The data are not publicly available due to the samples being commercially sensitive and the need for specialist processing of the data.

**Acknowledgments:** The authors are grateful to academic partners University of Sheffield and Cranfield University for the provision of the machined and WAAM samples, respectively. The authors also thank the ACEL and CfAM groups of the University of Nottingham for assistance in creating the laser-fabricated AM specimens.

**Conflicts of Interest:** The authors declare no conflict of interest.

### Abbreviations

The following abbreviations are used in this manuscript:

| | |
|---|---|
| SRAS | Spatial resolved acoustic spectroscopy |
| AM | Additive manufacturing |
| L-PBF | Laser powder bed fusion |
| WAAM | Wire arc additive manufacturing |
| WLAM | Wire laser additive manufacturing |
| SLS | Selective laser sintering |
| SAW | Surface acoustic wave |
| KED | Knife edge detector |
| SKED | Speckle knife edge detector |
| NDE | Non-destructive evaluation |
| XCT | X-ray computed tomography |
| EBSD | Electron back-scatter diffraction |
| SNR | Signal to noise ratio |

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
