# Peer review of "Imaging Microstructure on Optically Rough Surfaces Using Spatially Resolved Acoustic Spectroscopy"

_applsci, doi:10.3390/app13063424_

Round 1

Reviewer 1 Report

The authors present a broad experimental-based study of how surface roughness influences methods to image microstructure on optically rough surfaces.  The breadth of surface roughness types is commendable.  Further, the introduction of a definition of "surface roughness", and how that differs from form and waviness is useful (though, perhaps, the term wavy may be improved - perhaps "semi-periodic mesoscopic form" or periodic/pseudoperiodic form? - to be honest, I am not sure there is a particularly useful term, and I am sure the authors are striving for a concise term).  This prior comment is not a required correction, but asking "is there a better word".  Otherwise, Figure 1 is exceptionally useful.

In general, my comments fall into three categories: (i) minor edits (ii) more in-depth suggestions related to science or presentation and (iii) suggested aspects of their work that seems to have potential for subsequent impactful research.

(i) Regarding minor edits:

* Line 65: "is at its most efficient" is, perhaps, imprecise.  Perhaps the authors might consider changing this to "...firstly, additive manufacturing has many benefits that make it especially attractive when low production runs are required, even producing..."

* Line 92: "might have been through." - change to "might have experienced." to avoid ending in a preposition.

* Line 248:  Drop the word "state" - it isn't needed.

* Line 257:  There are two different step sizes given, which seems to be an error.  Please correct, or explain why both 100 x 200 um and 100 x 400 um are given.

* Lines 281-283:  Why is there such a variation in averages? This seems to beg for an explanation or clarity.  Please provide, if possible.  A footnote might be useful in this case, as the variability in these experiments is made at the time the experiments are conducted.

* Line 369: Honing and Turning should not be capitalized.

* Line 476:  "produced" should be "produce"

* Line 535:  extra space between et al. AND developed.  Please fix.

* Linee 563: "discontinuous" should be "discontinuities" 

(ii) Regarding more in-depth comments:

* Page 1, line 25.  The authors claim that the challenges associated with AM are more difficult as the materials state is "formed" during the manufacturing process.  The same statement can be made for any traditional manufacturing route.  Perhaps the authors mean that no other processing step affects the materials state - which isn't necessarily true (post deposition HT, HIP, or other strategies are adopted).  I might suggest that the authors consider revising this statement slightly, and point to the fact that materials state is influenced by both thermal gyrations and, owing to the near-net shape nature of AM parts, the materials state may vary by location within a part, due to the local differences in, e.g., heat extraction.

* Page 9, line 317: "There are a number of reasons for this...".  I might suggest the authors temper this very confident statement.  Perhaps "There are a number of possible reasons for this...".  From a technical perspective, I wonder why the authors expect that attenuation would result in light return dropping on the detector.  This series of hypotheses seems to be valid, but technically nuanced, especially for those not skilled in laser optics.  I might suggest that the paper might benefit with an additional figure (perhaps taking aspects of Fig. 1?) and provide a schematic (or series of schematics) to match this paragraph.  

* Paragraph starting on line 330.  This discussion is slightly more 'straight forward' than the previous paragraph.  However, the message lacks clarity.  I suggest that adding a sentence before "This change..." might help.  For example. "...ms-1.  There are two possible hypotheses that may expunge this change in wave speed.  The first may be related to either defocus of the generation patch, meaning the acoustic wave generation is inconsistent, or a dispersion of the SAW due to roughness.  Both are described, though the second seems more plausible. Regarding the former, for the roughest samples...".  The rest of the paragraph might need some slight edits, but a "bridging sentence" that pre-orients the reader to the more plausible explanation would be helpful.

* Figure 4(d) - the resolution of this figure in printed form makes me question the selection of the range of wave speeds.  If it is possible to change this figure, I would suggest that a wave speed of 2500-2800 may be clearer than 2400-2900.  I leave the decision to the authors, but suggest this might make the figure look better.  I also note that this picture of the comparator "washes out" towards the bottom right.  This means that for future citing articles/theses, the reproduction of this figure will be highly likely to "get worse".  Any way to improve this washing out now may have a payoff for the authors down the road.

* Figure 6:  How close to Gaussian is 6(d)?  Is this a qualitative or quantitative description.  I am ok with Gaussian, but might point that some readers might be interested in "how Gaussian" is it, especially in "2-D" (i.e., intensity profile along the x-axis vs intensity profile along the y-axis).  I expect this will be difficult to extract, and so I might suggest "In contrast, a Gaussian or nearly Gaussian reflection..." - or similar modification.

* Line 380-382, regarding color of surface of Ti.  I suspect that the difference in color has to do with the "time at temperature" differences between (a) and (b/c).  With the interpass rolling, the part cools off.  Thus, the total time at elevated temperature is reduced.  Conversely, without the interpass, the "thermal mass" is higher, and the part stays hotter.  In Ti, the color is directly related to the thickness of the oxide (or nitride).  The thinner, the less color.  Thickness is a function of time at temperature.  A minor technical detail, and certainly not the main subject of the paper, but it isn't correct to say "additional significant colouration of the surface [is] due to the heating and cooling of the sample".  Rather, "the coloration is a function of the thickness of the surface oxide/nitride, and the specimens with the interpass rolling have spent less time at elevated temperatures, cooling more between each additive layer".

* As I read the paper, I recognize that the data presented is of single orientation scans.  This should be made explicit (if not) in the methods part of the paper.  This explicit introduction early may help readers later (e.g., lines 433-437).

* Line 435-437 - this needs to be expanded.  I can visualize that this plane will have a higher degree of anisotropy in Cij, and thus, higher variation in wave speed.  Consequently, I agree with the interpretation by the authors.  However, not all readers will have thought through the fact that measurements of SAWs on a prism plane would be most strongly affected by the elastic anisotropy in Ti.  Expand (slightly) to clarify for the average reader.

* The authors should know that, for wire-based AM, the exact form of the waviness CAN actually be known.  Some companies have introduced methods to interpret this waviness during the manufacturing process.  Older (6 year old variants) methods use laser surface scanners between layers, providing adjustments to command the motion and avoid having "waviness-induced defects".  Newer variants use measurements of "focus/defocus".  I might suggest that the authors not that the waviness is difficult to know "a priori from the CAD file", but that emerging AM sensors mean that such waviness could be known during the AM process, or subsequent to the process using, for example, optical scanning methods"

(iii) Regarding areas to pursue:

* The line 392-394 is an especially interesting observation.  One wonders the cause.  Is it the result of a change in optical transmission of energy to the underlying material?  Or, is it is a capture of phonon by the transparent oxide, resulting in mode conversion into the oxide laterally?  Does it influence the elastic wave?  There may be some fascinating physics at play here.

* Figure 8 is an important figure.

* Lines 498-502:  I concur entirely, and this is an exciting suggestive paragraph.

The paper is quite nice, and I look forward to seeing it in the peer reviewed literature.  

Reviewer 2 Report

In this paper the challenges of using SRAS for inspection of real parts are discussed, especially from samples with waviness and roughness when inspecting parts from additive manufacturing.

The paper is well organized and understandable. It is scientifically sound and the conclusions drawn are well supported. With the detection it is a little sloppy: the presentation of the different principles and then only one method (Quartet) is used. Among the methods, two wave mixing (TWM) with fast crystals (as used e.g. by Tecnar and IOS) is totally missing and usually gives good results on rough surfaces. Why wasn't a TWM detector used as well? If this does not exist at your institute: An estimate would be fine if it would work here.

Reference to different detector types should be given e.g. in C. B. Scruby , L. E. Drain, Laser Ultrasonics Techniques and Applications (Adam Hilgar, Bristol, 1990)

Reference to TWM should be given e.g. in Bruno F. Pouet, Sridhar Krishnaswamy, Heterodyne interferometer with two-wave mixing in photorefractive crystals for ultrasound detection on rough surfaces, Appl. Phys. Lett. 69, 3782 (1996)

Reference to balanced TWM should be given e.g. in Hochreiner A, Reitinger B, Bouchal KD, Zamiri S, Burgholzer P, Berer T. Quasi-balanced two-wave mixing interferometer for remote ultrasound detection. J Mod Opt. 2013 Sep;60(15-16):1327-1331.

In detail:

- What excitation energy was used? Missing in chapter 2.2

- Chapter 3: Why are different detectors presented in chapter 2 and in chapter 3 only results with the Quartet are shown?

- Chapter 4 Figure 12: Which detector was used for the measurements?

- Chapter 4.3: The limits for SRAS are not only dependent on the excitation, but also on the detection. This should be described in addition.

- Chapter 4.4: What excitation energy is possible at 5kHz? Which average power results? Can thermal loading of the material and recrystallization (grain growth) occur? Especially with high necessity of averaging.

- The description of the individual detectors in chapter 2.2 is rather poor. Regarding the Quartet, three lines refer to Blum's paper. In the meantime, however, there is a fiber-based version of the Quartet (first version in free beam). There is also nothing more about the Polytec. In general, the focus distance and the spot size should still be included here. This provides information about the necessary maximum spot size and handling effort (measuring distance). 

- The description in Figure 2 is rather long for a and b; c to f are described only minimally. For example, what does the black and white image (Optical) tell us?

- Chapter 2.3 and Table 1: is the Spatial Resolution the diameter of the excitation beam? Or is

- Chapter 2.3.1: Spot diameter with the Quartet. Why was the Quartet used and not the SKED?

- Chapter 2.3.3: Why was it necessary to average 1000 times or more? What was the target SNR? Why is this the first time the Sked detector is used and not before?

Reviewer 3 Report

Dear Authors,

the paper is interesting and contains important research issues. In order to improve the article, I propose the following corrections.

1. In the abstract, why did the Authors put the word roughness in quotation marks?

2. The paper contains a general introduction to the subject, but there are no references to the literature, to the work of other authors on similar issues. Are there authors who have worked on similar issues? What are their conclusions?

3. What is the novelty of this paper? Please make this clear in your introduction and conclusions.

4. The introduction describes how to measure roughness with ultrasound. Please add references to what other roughness measurement methods are and how ultrasound compares to them. Advantages and disadvantages. Examples can be found in many papers, for example:

10.1007/BF00353195 – 3D optical microscopy, confocal microscopy

https://doi.org/10.3390/cryst11111371 - 3D optical microscopy, focus variation microscopy

https://doi.org/10.3390/coatings11050493 – 3D optical microscopy - interferometric

https://doi.org/10.3390/mi13101746 – stylus profilometer

5. In the introduction, it is also worth reviewing the literature on how to describe roughness. ISO standards, multi-scale analysis etc. The authors have indicated that they will describe these issues in section 2.1. However, in the review of the literature, writing at least briefly on this subject is missing.

6. Lines 107-110- these lines are unnecessary. The text should not be placed under the main chapter, but in the relevant subsections.

7. Lines 143-150 - There are no references to the literature and no indication of the authors who used these measurement techniques in their research.

8. Section 2.1 and 2.2. should be included in the introduction. Because in these sections there are no results of the authors' own research, and this is a literature review.

9. Section 2.3. Who is the supplier of research materials? Do the work objects have certificates confirming their chemical composition and material properties?

10. For the descriptions of the measuring equipment used, please provide the measurement accuracy.

11. In the test results, the surface texture is described by the Ra parameter. Please, at the initial stage of the paper, list other parameters characterizing the surface and indicate why Ra is the best choice for your research.

12. Lines 460-461 - What exactly is this small scale of roughness? Please use more numerical than descriptive values, because “small/large” are subjective terms.

13. A short conclusion section is missing at the end of the paper. The discussion section is presented, but it is worth summarizing the most important conclusions. This will clearly indicate the achievements of this paper.

Round 2

Reviewer 3 Report

The authors revised the manuscript with the required corrections. My questions are answered accordingly. I believe that the article in its current form can be published.